# Modeling of H5N1 influenza virus kinetics during dairy cattle infection suggests the timing of infectiousness

Oliver Eales[1,2]*, James M. McCaw[1,2], Freya M. Shearer[1,3]

1 Infectious Disease Dynamics Unit, Centre for Epidemiology and Biostatistics, Melbourne School of Population and Global Health, The University of Melbourne, Melbourne, Victoria, Australia, 2 School of Mathematics and Statistics, The University of Melbourne, Melbourne, Victoria, Australia, 3 Infectious Disease Ecology and Modelling, The Kids Research Institute, Perth, Western Australia, Australia

* oliver.eales@unimelb.edu.au

## Abstract

Since early-2024 unprecedented outbreaks of highly pathogenic avian influenza H5N1 clade 2.3.4.4b have been ongoing in dairy cattle in the United States with significant consequences for the dairy industry and public health. Estimation of key epidemiological parameters is required to support outbreak response, including predicting the likely effectiveness of interventions and testing strategies. Here, we pool limited publicly available data from four studies of naturally and experimentally infected dairy cattle. We quantify Ct value trajectories of infected dairy cattle and the relationship between Ct value and the log-titer of infectious virus, a proxy for infectiousness. We estimate that following infection minimum Ct values are rapidly reached within 1–2 days with a population mean Ct value of 15.7 (12.9, 18.4). We identify a threshold Ct value of 21.8 (19.9, 24.6), with values of Ct value above this threshold representing little-to-no infectious viral load. Finally, assuming a direct relationship between Ct value and infectiousness, we estimate the distribution of the duration of infectiousness for dairy cattle (i.e., the duration their Ct value remains below the critical threshold) with a population median of 7.8 (4.1, 13.9) days. Our estimates will be critical inputs to the development of outbreak management guide-lines and modeling analyses informing response strategies.

## Introduction

Since early-2024 unprecedented outbreaks of highly pathogenic avian influenza H5N1 clade 2.3.4.4b (genotype B3.13) have been ongoing in dairy cattle in the United States, following its introduction from wild birds [1]. As of 17 November 2025, 1,083 cattle herd outbreaks have been confirmed across 18 states [2]. These outbreaks have significant consequences for the dairy industry and public health. Reduced milk output of infected cattle, among other clinical symptoms [3], is resulting

**Data availability statement:** The data and code required to reproduce these analyses are available at https://github.com/Eales96/H5N1_viral_kinetics (Zenodo DOI: https://doi.org/10.5281/zenodo.17604863).

**Funding:** O.E. is supported by a University of Melbourne (https://www.unimelb.edu.au/) McKenzie Fellowship (503965). F.M.S. is supported by the National Health and Medical Research Council of Australia (https://www.nhmrc.gov.au/) through the Investigator Grant Scheme (Emerging Leader Fellowship, 2021/GNT2010051). J.M.M. is supported by the Australian Research Council (https://www.arc.gov.au/) through the Laureate Fellowship Scheme (FL240100126). This research is supported by the Australian Consortium of Epidemic Forecasting and Analytics (ACEFA), a National Health and Medical Research Council of Australia Centre of Research Excellence (2035303). F.M.S. and J.M.M.'s research is also supported by an Australian Research Council Discovery Project Grant (DP240102286). The funders had no role in study design, data collection and analysis, decision to publish, or preparation of the manuscript.

**Competing interests:** The authors have declared that no competing interests exist.

**Abbreviations:** Ct, cycle threshold; USDA, United States Department of Agriculture's.

in economic loss to the dairy industry. Spillover infections in dairy farm workers and raw milk contamination highlight the increasing public health risks of H5N1 clade 2.3.4.4b (genotype B3.13). Additionally, there is concern that continued circulation of H5N1 strains between cattle and other mammals may result in the virus adapting for human-to-human transmission [4], increasing the risk of a future influenza pandemic. To minimize these risks it is critical that outbreaks are: detected early through effective testing strategies; and rapidly contained through the implementation of effective outbreak controls.

Information on key epidemiological parameters is required to inform effective testing and outbreak control strategies. For example, the duration of infectiousness, the timing of the onset of infectiousness relative to clinical symptoms, and the timing of infection detection and isolation, all strongly influence the effectiveness of case isolation in controlling an outbreak [5]. Additionally, the effectiveness of testing of pooled samples—a method for testing entire dairy herds with limited resources—will strongly depend on the sensitivity of a test for detecting viral RNA, within-host viral kinetics, and the type of sample collected (e.g., blood, serum, and urine). Previous studies have suggested that milk and milking procedures are the dominant route of cow-to-cow transmission, with the highest viral loads detected in milk samples from infected cattle [3]. Thus, quantifying the viral kinetics of H5N1 2.3.4.4b (genotype B3.13) in milk samples of infected dairy cattle can inform testing and other outbreak control strategies.

Here we have pooled the limited publicly available data from four recently published studies [3,6–8] and performed statistical analyses to quantify H5N1 viral kinetics in milk samples of infected dairy cattle. We estimate the trajectories of cycle threshold (Ct) values as measured through rt-PCR testing of milk samples (a proxy for viral load), the relationship between Ct value and the log-titer of infectious virus (a proxy for infectiousness), and the duration of infectiousness for dairy cattle (i.e., the duration that their Ct value remains below a critical threshold). We interpret our findings, with respect to their implications for testing strategies and controllability of outbreaks in dairy cattle.

## Quantifying Ct value trajectories

We developed a Bayesian hierarchical model for quantifying Ct value trajectories in the milk of infected dairy cattle (see Methods for full description), building on models used to infer Ct value trajectories of SARS-CoV-2 infections during the COVID-19 pandemic [9,10]. Ct values are modeled as a function of time since infection, with parameters describing the peak viral load (i.e., the minimum Ct value reached, as low Ct values correspond to high viral loads), the time since infection at which the peak viral load is reached, and two rate parameters describing the initial viral growth rate before the peak (decline in Ct value), and the viral decay rate following the peak (increase in Ct value). We estimate population mean parameters and variation in these parameters between individuals (standard deviation in the mean parameters). The model is fit to longitudinal data from seven experimentally infected dairy cattle (Fig 1a) with daily rt-PCR testing of milk samples [6,7], and 12 naturally infected dairy

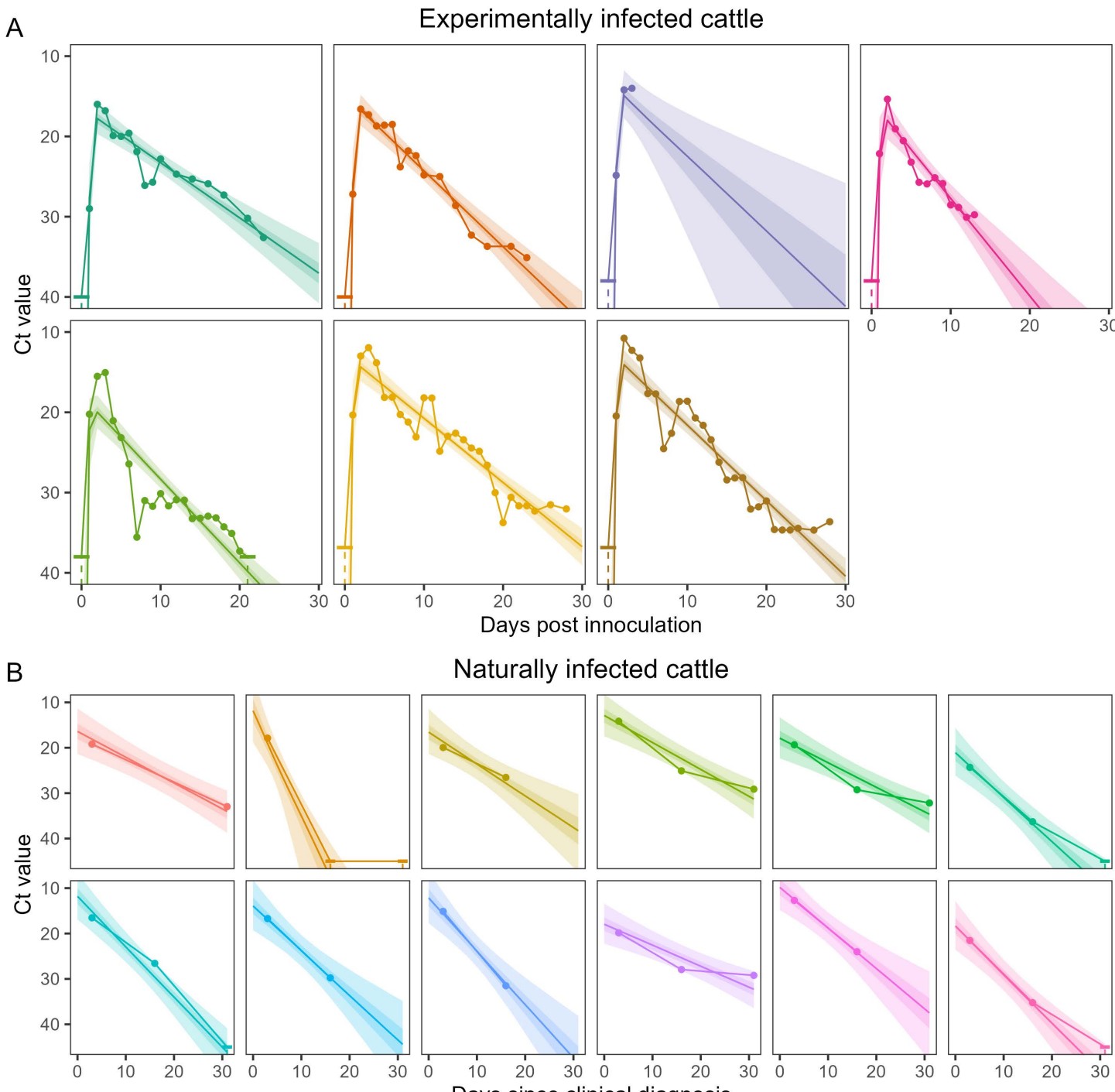

**Fig 1. Ct value trajectory model posterior fit to data.** Ct value trajectory model posterior estimates of the individual-level Ct value trajectories for experimentally infected cattle **(A)** and naturally infected cattle **(B)**. Ct value trajectory posterior estimates are shown with median (line), and 50% (dark shaded region) and 95% (light shaded region) credible intervals. Data points are connected via solid lines, with uncensored data shown as solid points, and censored data shown as a horizontal dash with a dashed line extending to the possible range of values. The data of the first two (dark green, orange) experimentally infected cows were from Baker and colleagues 2024 [7], the data of the next three (purple, pink, and light green) experimentally infected cows were from Halwe and colleagues 2024 [6], and the final two (yellow, brown) experimentally infected cows were from Facciuolo and colleagues 2025 [8]. The data for naturally infected cows were from Caserta and colleagues 2024 [3]. The data underlying this figure can be found in S1 Data and at http://dx.doi.org/10.5281/zenodo.17604863.

cattle (Fig 1b) who were tested on days 3, 16, and 31 post clinical diagnosis [3]. We assumed that the naturally infected dairy cattle were already in a phase of viral decay.

For this small sample of naturally and experimentally infected dairy cattle, we estimate that following infection minimum Ct values in milk are rapidly reached within 1–2 days with a population mean of 1.10 (1.03, 1.20) days (Figs 2a, S1, S2). We estimate a population mean in minimum Ct value of 15.7 (12.9, 18.4), followed by a gradual decline in viral load, with a population mean increase in Ct value of 0.88 (0.69, 1.13) per day. Estimates of the variation in parameters between individuals were uncertain, reflecting the small sample size of animals for which an entire Ct trajectory was available. However, there was greater certainty in our estimates for the variation in the viral decay rate, due to the additional data describing natural infections. Estimated variability in the viral decay rate suggests heterogeneity in the duration that an animal's viral load remains elevated (Fig 2A). We performed a sensitivity analysis fitting the model to subsets of the data (see Methods) and found no evidence of different parameter values between experimentally and naturally infected cattle (S1 Fig), although there were (as expected) higher levels of uncertainty in estimates.

## Relationship between Ct value and infectious virus

We inferred the relationship between Ct value and the log-titer of infectious virus in milk in infected dairy cattle by fitting a population-level statistical model describing a sigmoidal function to a dataset of infected cattle for which both

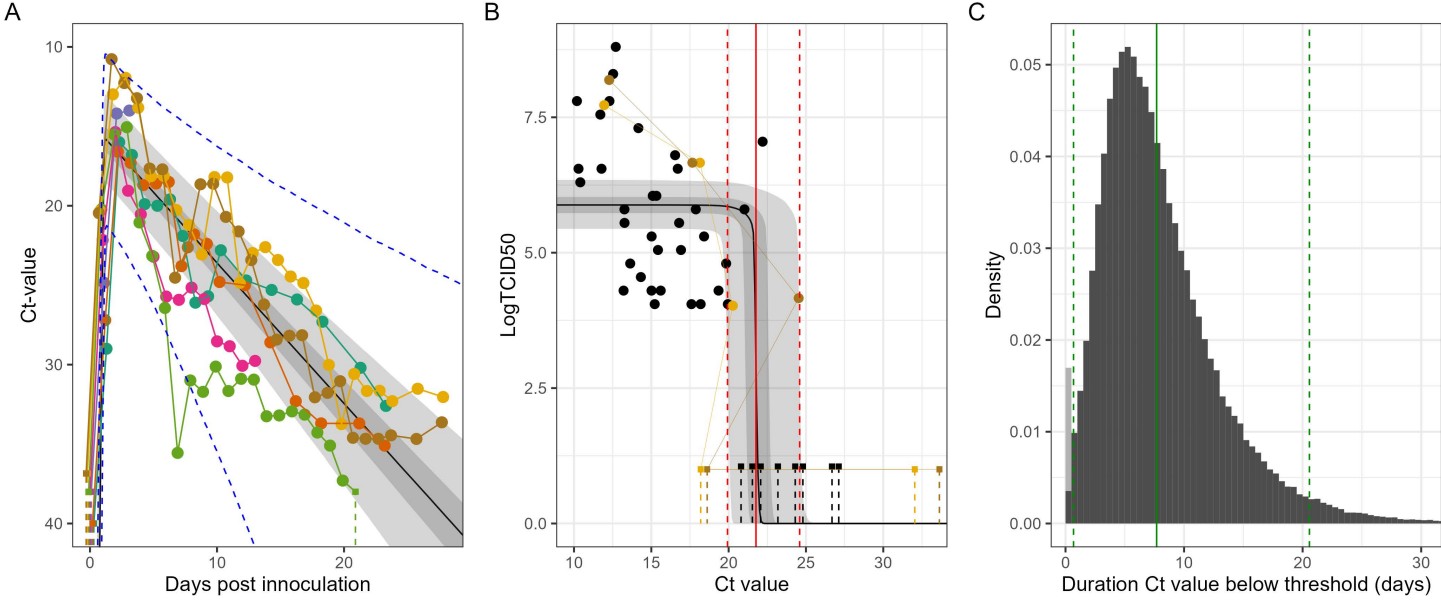

**Fig 2. Population distribution of Ct value trajectories and infectiousness. (A)** Ct value trajectory model posterior estimates of the population mean Ct value trajectory (i.e., using population mean parameters). Posterior estimates are shown with median (line), and 50% (dark shaded region) and 95% (light shaded region) credible intervals. Longitudinal data (points connected by solid lines) for the experimentally infected cows [6,7] are also shown (colors match Fig 1) with censored data represented by solid squares with a dashed line extending to the possible range of values. The central 95% interval (dashed blue lines) of the estimated population distribution of Ct value trajectories using the median of the population parameters (means and standard deviations) is also shown. **(B)** Posterior estimates of the relationship between log-titer of infectious virus and Ct value are shown with median (line), and 50% (dark shaded region) and 95% (light shaded region) credible intervals. Uncensored data are shown as solid points, and censored data are shown as solid squares with a dashed line extending to the possible range of values. Longitudinal data from Facciuolo and colleagues 2025 [8] $\beta_2$ in Methods). **(C)** The distribution of the duration that individual Ct value trajectories remain below the critical threshold Ct value (when they are likely infectious) estimated using: median population parameters (means and standard deviations) from the Ct value trajectory model (panel A); and the median threshold Ct value (solid red line in panel B). Also shown is the mean of the distribution (solid green line) and central 95% interval of the distribution (dashed green lines). The portion of the distribution that has a value of 0 is shaded light gray. The data underlying this figure can be found in S1 Data and at [http://dx.doi.org/10.5281/zenodo.17604863](http://dx.doi.org/10.5281/zenodo.17604863).

[https://doi.org/10.1371/journal.pbio.3003586.g002](https://doi.org/10.1371/journal.pbio.3003586.g002)

measurements were available. As Ct value decreases—representing a higher viral load—the log-titer of infectious virus increases (Fig 2b). Our estimates suggest that this increase occurs rapidly over a small range of Ct values centered on a critical threshold value of 21.8 (19.9, 24.6), with Ct values above this threshold representing little-to-no infectious viral load. Note that only limited longitudinal data were available describing Ct value and log-titer of infectious virus for infected cattle over time, and so a hierarchical approach to this part of the analysis could not be taken. Hence, our estimates can not account for variation between animals in the relationship between Ct values and log-titer of infectious virus.

## Duration of infectiousness

The dominant route of cow-to-cow transmission is thought to be through infectious virus in an infected animal's milk leading to direct infection of the mammary gland of another animal, either due to contamination of the environment (e.g., floor and bedding) or contamination of milking apparatus [3,11,12]. The log-titer of infectious virus in milk is thus a useful proxy for estimating the infectiousness of an animal. We have estimated population mean and variation in Ct value trajectories, and the relationship between Ct value and log-titer of infectious virus, under various modeling assumptions. We combine these estimates to infer the duration in which infected cattle have a Ct value below the critical threshold value (when they are likely infectious).

Accounting for parameter uncertainty in both models, we estimate that the mean duration of infectiousness of an infected animal is 7.8 (4.1, 13.9) days (Fig 2C). This estimate is sensitive to the critical threshold Ct value, with a higher critical threshold Ct value leading to a longer mean duration of infectiousness (S3 Fig). We estimate that there is variation in the duration of infectiousness between animals with median parameter estimates (Fig 2C) suggesting that 1.3% of animals do not become infectious, and that 2.5% of animals are infectious for greater than 20.6 days. We estimate that 95% of cattle will no longer be infectious from 18.8 (11.8, 36.1) days post infection, which has implications for determining the appropriate policy for isolation duration. However, the estimated tail values of this infectiousness duration will be highly sensitive to the assumed distribution and so, these estimates should be interpreted with caution.

## Effectiveness of isolating infected cattle

Using the full posterior for our estimates of the duration of infectiousness (population distribution), we estimated the proportion of transmission events that would be prevented given case isolation (Figs 3, S4)—a standard practice for preventing transmission of other pathogens affecting dairy cattle [13]. We calculate this reduction in transmission for different proportions of infections that are isolated (0–1) and at different times post infection (0–15 days). We estimated high uncertainty in the effectiveness of case isolation. For example, if 100% of cases were isolated one week post infection we estimated 38% (17%, 61%) of onward transmission would be prevented. As expected, the longer the estimated duration of infectiousness, the more effective we estimated case isolation to be. This is because longer infectious periods (for the same overall transmissibility and timing of isolation) provide a longer time window for the prevention of onward transmission events.

The effectiveness of case isolation for preventing onward transmission depends on: the proportion of infections that can be detected and isolated; and the relative timing of detection to infectiousness (Fig 3). These two factors are dependent on the rate and timing of clinical symptoms in infected animals, and the ability for symptoms to be recognized (e.g., reduced feed intake and rumination time may be difficult to identify). In some experimentally infected cattle, changes to milk color and consistency began as early as two days post inoculation [7], suggesting that careful monitoring of milk output during an outbreak may allow early detection of infections. However, it is currently not clear what proportion of naturally infected cattle exhibit identifiable symptoms, if the timing is comparable to experimentally infected cattle, nor if infectiousness is correlated with the severity (and therefore identifiability) of symptoms. Epidemiological investigations of the first H5N1 outbreaks on dairy farms reported attack rates of clinically affected cattle ranging from 5% to 23% [3], but it is unclear what the underlying infection attack rates were for these outbreaks (i.e., including both clinical and subclinical

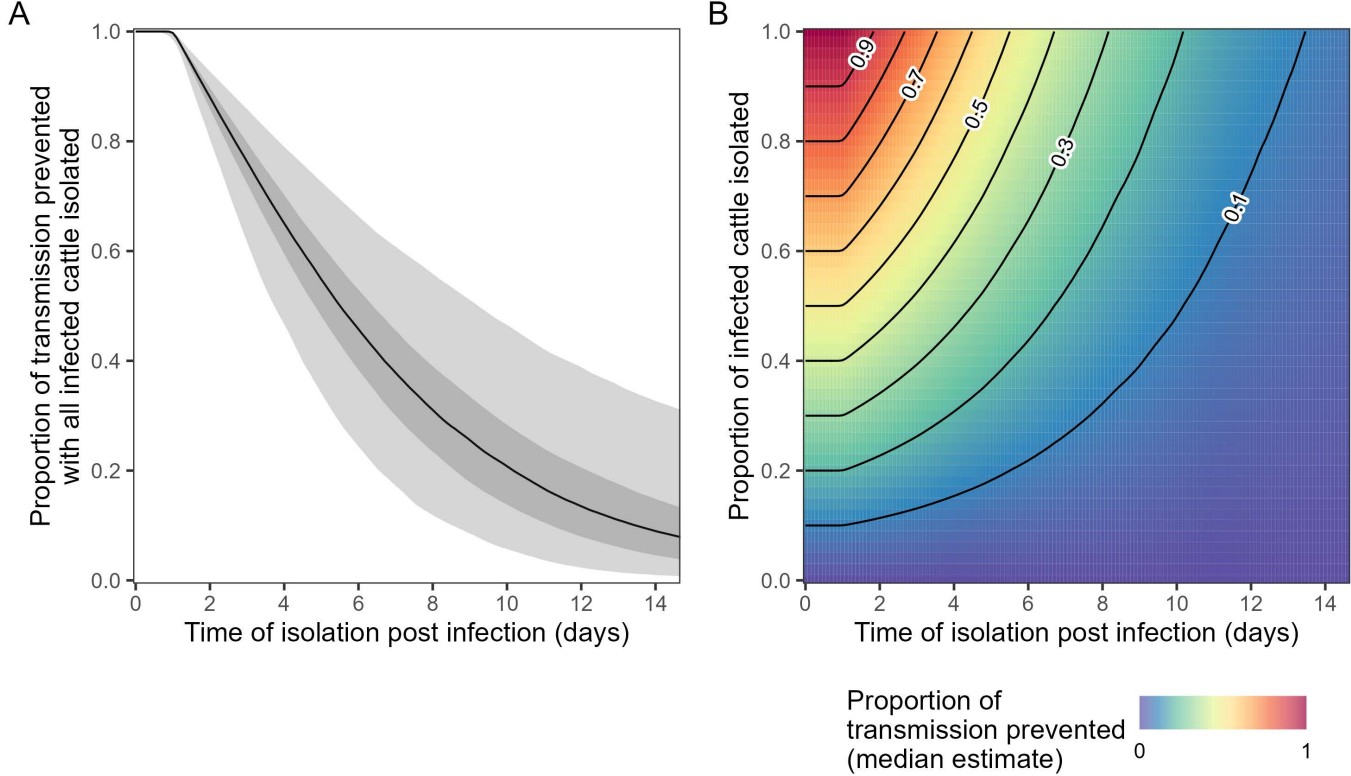

**Fig 3. Proportion of transmission prevented by case isolation. (A)** The proportion of transmission prevented, if all infected cattle are isolated, as a function of the time post infection at which infected cattle are isolated (x-axis). Posterior estimates are shown with median (black line), and 50% (dark shaded region) and 95% (light shaded region) credible intervals. **(B)** The proportion of transmission events prevented (color) as a function of: the time post infection at which infected cattle are isolated (x-axis); and the proportion of infected cattle that are isolated (y-axis). The proportion of transmission events prevented is calculated across the entire posterior distribution for both models; plotted is only the median value (50 percentile panel). The central 50% and 95% credible intervals for all x- and y-value are shown in S4 Fig, and for the case when 100% of infected cattle are isolated, in panel A. Contour lines (proportion of transmission prevented: 0.1–0.9 in intervals of 0.1) connect points with the same value (black lines). The data underlying this figure can be found in S1 Data.

infections). A recent seroprevalence study, following an outbreak of H5N1 on a dairy farm, estimated that ~75% of infections were subclinical [14]; although this value will be sensitive to the definition of clinical infection used (i.e., severity/identifiability of the symptoms used to define a clinical case). Our results indicate that case isolation is unlikely to be effective at reducing transmission if infected animals enter isolation beyond seven days post infection even if a very high proportion (e.g., 90%–100%) of infections can be detected and isolated. Furthermore, case isolation within 7 days of infection is only likely to be effective if the true duration of infectiousness is within the upper bound of our estimated distribution (e.g., 95% CrI of mean duration of infectiousness is 4.1–13.9 days). In general, higher proportions of and faster rates of infection detection will result in greater effectiveness of case isolation. However, it will be difficult for high proportions of infections to be detected if a high fraction of infections are subclinical, as has recently been estimated [14].

## Pooled testing of milk vats

Timely detection of H5N1 outbreaks in dairy farms is crucial for minimizing transmission within a farm and the risk of spread between farms. Early detection can allow rapid implementation of control measures such as case isolation, quarantine, livestock transport controls, and enhanced cleaning protocols. While it would likely be impractical to frequently test

all cattle—or even all clinically affected cattle—pooled testing of samples from multiple cattle could be feasible and effective at identifying outbreaks. However, the utility of pooled testing depends on the effectiveness of the test at identifying an infection in a diluted sample (i.e., diluted by many uninfected samples). Pooled testing of milk samples is an obvious strategy to employ for detection of H5N1 outbreaks in dairy farms: the highest Ct values have consistently been measured in milk samples of infected animals; daily (at least) milking is routine; and milk samples for a single dairy farm are often pooled into a central milk vat.

We calculated the expected Ct value of a pooled milk sample containing the milk from a single infected animal, varying the: Ct value of the milk of the infected animal if tested in isolation; the number of milk samples in the pooled sample; and the efficiency of the rt-PCR test used to measure the Ct value (Fig 4). The ability of the pooled test to identify the presence of a single infected animal will depend on the relative value of the expected Ct value and the limit of detection of the test (the number of cycles performed). The lower the expected Ct value relative to the limit of detection, the greater the probability of detection. For a milk sample from one animal with a Ct value of 15.7 (the population mean peak Ct value), assuming no reduction in milk output, the expected Ct value in a milk sample from 40,000 cattle was estimated to be 32.2 (90% rt-PCR test efficiency), 31.6 (95% rt-PCR test efficiency), and 31.0 (100% efficiency). These values are all well

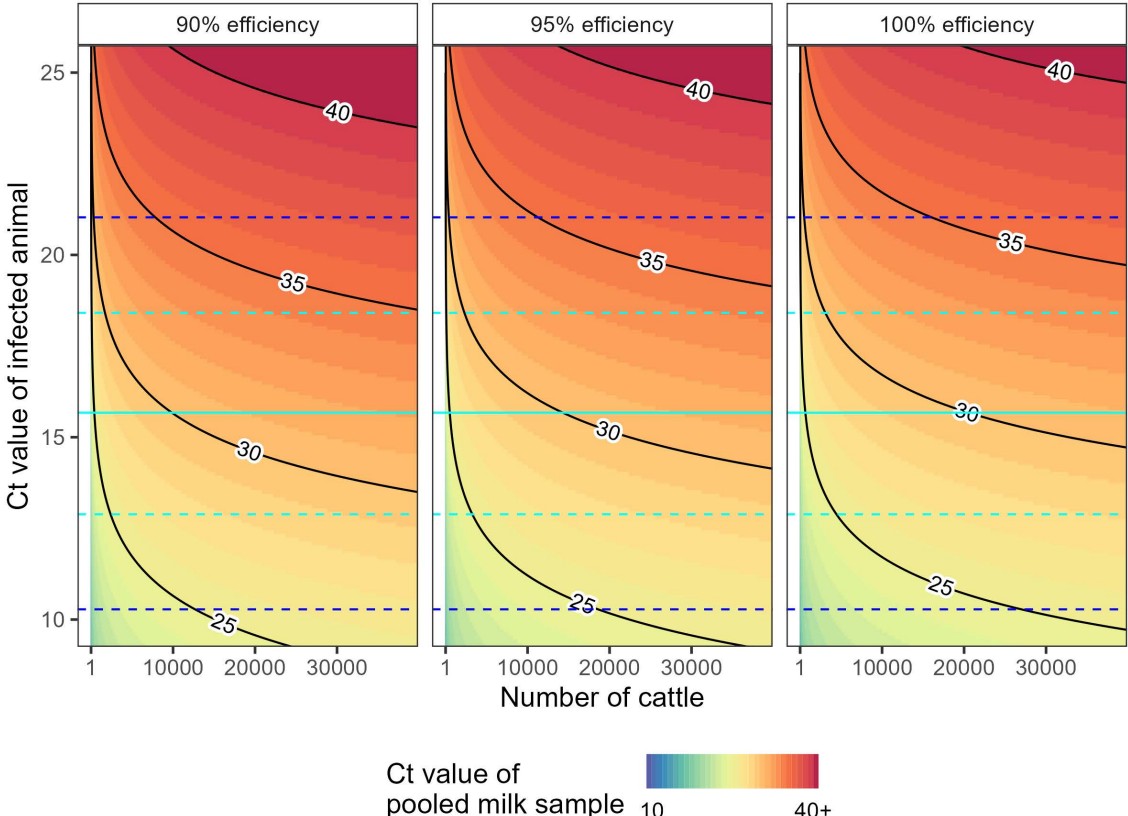

**Fig 4. Expected Ct values of pooled milk samples.** Expected Ct value of a pooled milk sample (colour) as a function of: the Ct value of the infected animal (y-axis, one infected animal contributing to the pooled sample); the total number of cattle contributing to the sample (x-axis); and the efficiency of the rt-PCR test used (panels). Note that an rt-PCR test of 100% efficiency doubles the viral concentration with each round of amplification (Ct value increasing by 1). Contour lines (Ct values: 25, 30, 35, and 40) connect points on the graph with the same value (labelled black lines). The median (solid line) and 95% credible interval (dashed lines) for the population mean minimum Ct value ($\mu_P$) for a single infected animal is highlighted (cyan). The 95% credible interval (dashed lines) for the population distribution of minimum Ct values using mean population parameters, $\mathbf{N}(\mu_P, \sigma_P)$, for a single infected animal is also highlighted (blue). The data underlying this figure can be found in S2 Data.

within realistic limits of detection (e.g., 38 cycles [6], 40 cycles [7], 45 cycles [3]). Even at higher Ct values—either due to an infection with lower viral load or a sample time not at the peak—the expected Ct value is likely within realistic limits of detection, particularly for test efficiencies of 90% or more [15]. These estimates assume that an infected animal produces the same milk output as healthy cattle and thus are likely an overestimate of the sensitivity of pooled testing. However, even when assuming a reduced milk output for infected animals as low as 10% of an uninfected animal (S5 Fig), we found that a typical Ct value (population mean peak Ct value) was likely to be within realistic limits of detection. The expected Ct values would decrease with a greater number of infected cattle making a positive detection even more likely. This suggests that frequent (e.g., weekly) pooled testing of samples from dairy farm milk vats would rapidly identify outbreaks, with a single index case detectable even on large dairy farms (e.g., 30,000–40,000 cattle).

## Limitations

There may be issues with the representativeness of the data, given the small number of animals sampled and the lack of data from naturally infected animals. While our analysis found no evidence for a difference in the viral decay rates between experimentally and naturally infected cattle, the only longitudinal data that measured Ct values over the entire course of infection were from experimentally infected cattle; viral kinetics of H5N1 infection in naturally infected cattle may differ. For example, naturally infected cattle may have been initially exposed to a lower volume of infectious virus than experimentally infected cattle (a high volume of infectious virus is often used during experimental inoculation to ensure infection), which could result in a longer time until the minimum Ct value is reached [16]. Indeed, one study that inoculated four cows with a significantly lower volume of infectious virus (than the experimental infection studies whose data were analyzed in this paper) and found a longer duration to peak in viral loads [17]. Differences in the time to minimum Ct value would alter the period for which we estimate that cattle remain infectious (assuming the same rate of viral decay) and thus alter our estimates of the effectiveness of control measures such as case isolation. For example, if the time to peak viral load (minimum Ct value) was longer in naturally infected cattle (e.g., as reported in Lee and colleagues [17]), then we might expect a longer delay between infection and an animal becoming infectious, increasing the effectiveness of case isolation (more cattle isolated earlier in their infectious period). Additionally, the data we had from naturally infected cattle were identified through early outbreak investigations, which may have been preferentially biased towards more symptomatic or severe cases. This underscores the need for more detailed studies aimed at capturing the full course of infection in naturally infected cattle, with unbiased selection of cattle—longitudinal sampling of randomly selected cattle on a dairy farm during an outbreak could provide such data.

Due to the limited data available, we could only implement simple models describing the viral kinetics of H5N1 infection in dairy cattle. We assumed a single parameter described the viral decay rate following the peak in Ct value. However, there may be biphasic decay (e.g., as observed in other virological [18,19] and serological [20,21] datasets) with an initially high decay rate followed by a lower long-term decay rate. This may have implications for our estimates of the duration of infectiousness (and effectiveness of case isolation) as a faster decay rate rate could lead to a reduction in the time Ct values remained below the critical threshold value. Additionally, the lower decay rate and the possibility of longer term shedding of virus may have implications for testing strategies—long-term shedding could lead to positive detection for a period of time following the end of an outbreak that would need to be considered when developing interventions.

Our model assumes a simple relationship between Ct value and infectiousness (specifically log-titer of infectious virus)—an animal with a Ct value below a threshold is infectious and above a threshold is not infectious. However, the infectiousness profile of an individual animal may vary more smoothly over time. The exact relationship would be difficult to quantify and would depend on: (1) the infectious viral load over time; and (2) the relationship between infectious viral load and infectiousness. We assumed a simple relationship between Ct value and the log-titer of infectious virus, identifying a threshold Ct value above which there was little-to-no infectious virus load. Our estimate of the duration of infectiousness effectively assumes that: when the Ct value is below the threshold the infectious viral load is high (and constant) and

the cow is infectious (with a fixed probability of infecting milking apparatus (given contact) per unit time); and when the Ct value is above the threshold, the infectious viral load tends to zero and the cow is not infectious. However, it is possible that the infectious viral load (and so infectiousness) changes more smoothly with Ct value (particularly above and below the threshold value). Higher Ct values that we observe earlier in infection might indicate that cattle's infectiousness profile should be skewed to earlier times. Low but non zero infectious viral load later in the infection may skew the infectiousness profile to later times; a recent study showed that as little as 10 $TCID_{50}$/ml of virus (below, or at, the limit of detection of the data used in our analysis) was sufficient to establish infection in the bovine mammary gland [17].

We had limited longitudinal data describing the relationship between measured Ct values and the log-titer of infectious virus, and so could not fit a hierarchical model that allows for variation in the relationship between individuals in the population. Hence, our model assumed a population-level relationship between Ct values and the log-titer of infectious virus; as such the 'critical threshold' Ct value that we estimated represents a population average (for this specific sample of cattle, which may not be representative of the entire dairy cattle population). There may be individual-level variation in this relationship, with some cattle still yielding infectious virus when their Ct value has passed above this population-level threshold value (e.g., [6]). Individual-level variation could influence estimates of the distributions of the duration and timing of infectiousness.

Additionally, our modeling assumption of a constant relationship between Ct value and the log-titer of infectious virus, does not consider other possible nonlinear interactions with the host immune response. Halwe and colleagues [6] found that antibody levels increased from ~7 to 11 days post infection in two experimentally infected cattle, with neutralizing antibodies detected in milk samples from 9 days post infection. It stands to reason that the presence of neutralizing antibodies is likely to substantially reduce or eliminate the potential for onward transmission. The antibody response will (we expect) be induced by the presence of infectious virus; thus quantifying a relationship between Ct value and the log-titer of infectious virus (as we have done) can implicitly account for some of the effects of underlying (causally related) antibody dynamics (that reduce the infectious viral load). However, our statistical model assumed (by necessity) a simple relationship between Ct value and the log-titer of infectious virus, which may not be able to quantify a more complex relationship between Ct value, log-titer of infectious virus, and antibody response. Ideally, more longitudinal studies that measure Ct value trajectories would also concurrently collect data on the log-titer of infectious virus and antibody levels. Models may then be developed that quantify the (potentially nonlinear) relationships between these three quantities, while accounting for variation between individual cattle.

We considered the effectiveness of case isolation and pooled testing of milk samples in isolation but did not consider their combined effect. When pooled testing is conducted regularly, outbreaks on dairy farms may be detected rapidly, allowing for the timely implementation of interventions. Case isolation is commonly used for preventing transmission of diseases afflicting dairy cattle (e.g., mastitis [13]). We inferred the effect of case isolation on transmission, by estimating the proportion of transmission prevented as a function of the proportion of infections isolated and the time from infection to isolation. However, the effect of a transmission reducing intervention on the overall size of an outbreak will be highly dependent on the inherent transmissibility of the pathogen (i.e., the basic reproduction number, $R_0$: the expected number of secondary infections arising from a primary infection in an otherwise susceptible population), and how soon after the onset of the outbreak the intervention is implemented. The earlier interventions are implemented, the smaller the outbreak. The higher the basic reproduction number ($R_0 \gg 1$), the smaller the impact of interventions. Note that if $R_0$ is close to one, then only a small reduction in transmission would be required for an outbreak to be controlled (i.e., an immediate reduction in infections). However, this is unlikely to be the case for H5N1 in dairy cattle given the estimated attack rate on farms—one study found almost 90% of cattle were seropositive following an outbreak (suggesting an $R_0$ well above one). To fully quantify the effect of case isolation and frequent pooled testing on the size of H5N1 outbreaks on dairy farms would require a mathematical model describing the on-farm epidemiological transmission dynamics.

## Conclusions

We have inferred the viral kinetics of H5N1 infections in dairy cattle. While our statistical models have been designed to analyze currently available data, it will be important to update our estimates as the situation evolves and additional data are collected. Using our estimated Ct value trajectories for the milk of infected dairy cattle, we have demonstrated that pooled testing of milk samples would likely be highly sensitive at detecting the presence of a single infected animal on even the largest dairy farms (e.g., 30,000–40,000 cattle) or central processing facilities. This supports the implementation of the United States Department of Agriculture's (USDA) National Milk Testing Strategy [22], which includes: testing of samples from milk silos at national dairy processing facilities to determine the presence of H5N1 at the state level; and bulk tank sampling programs in (identified) affected states to identify dairy herds affected by H5N1. The testing strategy is designed to increase USDA's and public health partners' understanding of where the virus is present in the United States. It has been implemented in 46 states of the United States, allowing states and herds to be classified based on their H5N1 affected status, and hence interventions and resources to be better targeted [22]. Assuming a direct relationship between Ct value and infectiousness, we have estimated the timing and duration of infectiousness, and demonstrated that case isolation is unlikely to be effective for outbreak control if infected animals enter isolation beyond seven days post infection, even if a very high proportion (e.g., 90%–100%) of infections can be detected and isolated. It is unlikely that such a high proportion of infections can be detected, given recent findings suggesting that a high fraction of infections are subclinical [14]. This highlights the need for the development of novel on-farm intervention strategies that effectively reduce within-herd transmission. Our estimates will be useful inputs to the development of outbreak management guidelines and future modeling analyses required to inform outbreak response strategies.

## Methods

### Ct value data

We retrieved longitudinal data describing the Ct value in milk samples from five experimentally infected (H5N1 B3.13) dairy cattle (inoculated by intramammary route) [6,7], and 12 naturally infected dairy cattle [3]. All animals included in the study were reported to show clinical signs.

### Ct value data: experimentally infected cattle

The longitudinal data for two experimentally infected (H5N1 B3.13) dairy cattle are available from the supplementary material of Baker and colleagues 2024 [7]. The Ct values were from rt-PCR testing of milk samples in buckets. Ct values were measured daily up to 23 days post inoculation. The limit of detection was a Ct value of 40 (i.e., up to 40 rounds of amplification). The full study protocol is available in the published article [7].

The longitudinal data for three experimentally infected (H5N1 B3.13) dairy cattle were extracted from Halwe and colleagues 2024 [6]. The Ct values from rt-PCR testing of milk samples were not available in the Supplementary Materials, and so the values were digitally extracted from the original figure (Fig 3 Halwe and colleagues 2024 [6]) using PlotDigitizer [23]. This extraction process will have introduced a small amount of additional uncertainty into the Ct values. Ct values were measured daily up to 21 days post inoculation. Measurements for one animal were only available up to 3 days post inoculation, and for another up to 13 days post inoculation due to these animals meeting conditions for humane euthanasia. The limit of detection was a Ct value of 38 (i.e., up to 38 rounds of amplification). The full study protocol is available in the published article [6].

The longitudinal data for two experimentally infected (H5N1 B3.13) dairy cattle are available from the supplementary material of Facciuolo and colleagues 2025 [8]. The Ct values were from rt-PCR testing of milk samples from each individual teat, and were presented as $TCID_{50}$ equivalent/ml. The average across the four samples was calculated and then converted to a Ct value using the standard curve ($Ct = 44.843 - 1.736 \times ln(x)$, where $x$ is the average value of four milks

samples in $TCID_{50}$ equivalent/ml). Ct values were measured daily up to 28 days post inoculation. The limit of detection was 100 $TCID_{50}$ equivalent/ml, which corresponds to a Ct value of 36.85. The full study protocol is available in the published article [8].

### Ct value data: naturally infected cattle

The longitudinal data for the 12 naturally infected dairy cattle are available from the supplementary material of Caserta and colleagues 2024 [3]. Ct values were measured using rt-PCR testing of milk samples collected 3, 16, and 31 days post clinical diagnosis for a subset of cattle on-farm 3 of the study. Data were available on 15 animals, but we excluded three cattle for which there was either only one data point available or for which all measurements were outside the limit of detection. The limit of detection was a Ct value of 45 (i.e., up to 45 rounds of amplification). The full study protocol is available in the published article [3].

### Ct value trajectory model

We fit a Bayesian hierarchical model quantifying the Ct value trajectories in infected dairy cattle. For each experimentally infected dairy cow, $i$, Ct values are modelled as a function of time since infection, $t_{inf}$:

$$Ct_i(t_{inf}) \ = \ P_i - r_i \times (t_{inf} - \tau_i) \ \text{for } t_{inf} < \tau_i$$

$$Ct_i(t_{inf}) \ = \ P_i + d_i \times (t_{inf} - \tau_i) \ \text{for } t_{inf} \geq \tau_i$$

The four parameters are: $P_i$, the minimum Ct value reached; $\tau_i$, the time since infection at which the minimum is reached; $r_i$, the initial viral growth rate (decline in Ct value) before the minimum Ct value is reached; $d_i$, the viral decay rate (increase in Ct value) after the minimum Ct value is reached.
For each naturally infected dairy cow, $j$, Ct values are modelled as a function of time since clinical diagnosis, $t_{cd}$:

$$Ct_j(t_{cd}) \ = \ Ct_j(0) - d_j \times t_{cd}$$

The two parameters are: $Ct_j(0)$, the Ct value on the day of clinical diagnosis; and $d_j$, the viral decay rate (increase in Ct value) after the minimum Ct value is reached. This explicitly assumes that the naturally infected dairy cattle were all in the phase of viral decay when their first sample was collected (3 days post clinical diagnosis). This is consistent with our estimates of $\tau_i$ which was 1–2 days post inoculation in experimentally infected cattle. Note that clinical diagnosis will be later than the date of infection as well.

The model was implemented in STAN [24] and parameter posteriors were sampled using a No-U-Turns Sampler [25] fitting to all data (sensitivity analysis described below). The model was run for 4000 iterations across four chains with a burn in period of 1000 iterations. The prior distributions used for all parameters are described in S1 Table. We assumed that uncensored Ct value data (not at the limit of detection) was normally distributed around modelled Ct values with standard deviation, $\sigma_{Ct}$. Ct value data at the limit of detection were treated as additional parameters with constant priors (outside the limit of detection) and were also assumed to be normally distributed around modelled Ct values with standard deviation, $\sigma_{Ct}$.

The individual-level parameters describing the Ct value on the day of clinical diagnosis for the naturally infected cattle, $Ct_j(0)$, were given an uninformative constant prior (with a minimum possible value of 0). All other individual-level parameters were assumed to follow a population-level distribution with parameters describing their mean, $\mu$ and standard deviation, $\sigma$:

$$P_i \sim N(\mu_P, \ \sigma_P)$$

$$\tau_i \sim N(\mu_\tau, \ \sigma_\tau)$$

$$log(r_i) \sim N(\mu_r, \ \sigma_r)$$

$$log(d_i) \sim N(log(\mu_d), \ \sigma_d)$$

$$log(d_j) \sim N(log(\mu_d), \ \sigma_d)$$

The rate parameters were assumed to be log-normally distributed while the other parameters were assumed to be normally distributed. All individual-level parameters, and population mean and standard deviation parameters, were given uninformative constant priors. Individual-level and population mean viral decay and growth rate parameters were assumed to be in the range 0.01 to 100 per day. Individual-level and population mean minimum Ct value parameters were assumed to be greater than 0. Individual-level and population mean time until minimum Ct value parameters were assumed to be in the range 0–10 days post inoculation.

## Sensitivity analyses

We ran additional models as a sensitivity analysis (S1 Fig). We fit the model to data from only either experimentally infected cattle or naturally infected cattle (note that for the latter only the viral decay rate parameters can be fit using such data).

Additionally, reflecting that assays performed in different labs may have a systematic difference in quantitative performance, we fit the model to all data, but included a dataset-level coefficient in the observation model that converted Ct value from one study to another (i.e., allowing the same Ct values in different datasets to reflect different viral loads). Note that this is already intrinsically included for natural infections versus experimental infections in all models because we only consider the change in Ct value for naturally infected cattle (i.e. the decay rate parameter). We include two additional parameters ($Ct_{adj,k}$, where k is the number of datasets minus one), allowing adjustment in Ct value between the three H5N1 challenge studies. The Ct value model was defined in terms of the experimentally challenged cows from Baker and colleagues [7] (i.e., Ct value model is unchanged for the Baker and colleagues dataset). The Ct value for experimentally challenged cows in the Halwe and colleagues [6] and Facciuolo and colleagues [8] dataset was then given by:

$$Ct_i(t_{inf}) = Ct_{adj,k} + P_i - r_i \times (t_{inf} - \tau_i) \text{ for } t_{inf} < \tau_i$$

$$Ct_i(t_{inf}) = Ct_{adj,k} + P_i + d_i \times (t_{inf} - \tau_i) \text{ for } t_{inf} \geq \tau_i$$

where the adjustment parameter, $Ct_{adj,k}$, was different for each of the two datasets. The Ct value model for the naturally infected cows was unchanged.

Finally, we fit the model to all data, but assumed that the viral decay rate parameters for experimentally ('exp inf') and naturally ('nat inf') infected cattle came from distinct distributions:

$$log(d_i) \sim N(log(\mu_{d, \ exp \ inf}), \ \sigma_{d, \ exp \ inf})$$

$$log(d_j) \sim N(log(\mu_{d, \ nat \ inf}), \ \sigma_{d, \ nat \ inf})$$

## Pooled testing analysis

We calculated the expected Ct value that would be measured when testing pooled samples of milk in which a single positive milk sample was present. We calculated the expected Ct value of a pooled sample using the equation:

$$Ct_{Pooled} = Ct_{inf} + Log_{1+\epsilon}\left(1 + N - 1/m\right).$$

Here, $Ct_{inf}$ is the expected Ct value of the infected animal's milk sample tested in isolation, $\epsilon$ is the PCR efficiency of the rt-PCR test [15] $N$ is the total number of cattle in the sample ($N-1$ are uninfected), and $m$ is the amount of milk an infected animal produces relative to an uninfected animal. In the main analyses we assume $m = 1$, which corresponds to the volume of milk from each cow present in the sample is equal, which may not always be the case as some infected cattle have been reported to have reduced milk output. As a sensitivity analysis we also assume $m = 0.5$, $m = 0.25$, and $m = 0.1$, corresponding to infected animals producing 50%, 25%, and 10% as much milk as uninfected animals, respectively. This equation assumes that when the number of cattle in the pooled sample doubles the viral concentration halves, and so more rounds of amplification are required to detect the virus. Note that when the efficiency of the rt-PCR test is 100% then doubling the number of cattle means an additional round of amplification is required (i.e., the expected Ct value increases by 1).

## Ct value and infectious virus log-titer model

We fit a statistical model to quantify the relationship between Ct value and the log-titer of infectious virus in milk samples.

Data on measurements of Ct value and the log-titer of infectious virus from milk samples of infected cattle were available from the supplementary material of Caserta and colleagues 2024 [3]. Measurements of Ct value (Fig 1a Caserta and colleagues 2024 [3]) and log-titer of infectious virus (Fig 1f Caserta and colleagues 2024 [3] were matched by sample IDs (see Data and Code availability). In total we were able to identify 43 pairs of measurements. The log-titres of infectious virus were expressed as the $log_{10}$ of the 50% tissue culture infectious dose ($TCID_{50}$) per millilitre ($TCID_{50}ml^{-1}$). The limit of detection was $10^{1.05} TCID_{50}ml^{-1}$. The full study protocol is available in the published article [3].

Additional data on measurements of Ct value and log-titer of infectious virus from milk samples of two experimentally infected cattle were available from the supplementary material of Facciuolo and colleagues 2025 [8]. The Ct value of the milk samples were the same as described above (Section "Ct value data: experimentally infected cattle"), the log-titre of infectious virus was taken as the average of four measurements (one made for a milk sample from each teat). Measurements were made at 3, 5, 7, and 10 days post inoculation for each cow (8 pairs of measurements). The limit of detection was $10^{1.0} TCID_{50}ml^{-1}$.

We modelled the log-titre ($Log_{10}(TCID_{50}ml^{-1})$) of infectious virus, $LT$, as a sigmoidal function of the Ct value:

$$LT = \frac{\beta_0}{1 + exp(\beta_1 \times (\beta_2 - Ct))}$$

where $\beta_0$ describes the maximum value of $LT$. $\beta_1$ describes the steepness of $LT's$ transition from 0 to its maximum value of $\beta_0$. $\beta_2$ describes the Ct value at which $LT$ reaches half of its maximum value. When fitting the model (see below) the posterior distribution for $\beta_1$ only allows for rapid transitions from 0 to the maximum value of LT (i.e. large values of $\beta_1$); the parameter $\beta_2$ thus represents a critical threshold Ct value (referred to as such in the main text) above which the log-titre of infectious virus is zero, and below which the log-titre of infectious virus is at its maximum.

The model was implemented in STAN [24] and parameter posteriors were sampled using a No-U-Turns Sampler [25] fitting to all data (sensitivity analysis described below). The model was run for 10,000 iterations across four chains with a burn in period of 2,000 iterations. We allowed for uncertainty in both measured Ct values and log-titres of infectious virus

by assuming modelled $LT$ was a function of modelled Ct values, and treating modelled Ct values (each corresponding to a data point) as additional parameters with uninformative constant prior distributions (limited to be positive in value). For the eight points in which the log-titre of infectious virus was at the limit of detection, we treated the true value of the log-titre as an additional parameter with a constant prior distribution and upper limit of 1.05 $Log_{10} TCID_{50} ml^{-1}$ (the limit of detection). We assumed that Ct value data were normally distributed around the corresponding modelled Ct value with standard deviation, $\sigma_{Ct}$. We assumed that log-titre data were normally distributed around the corresponding modelled $LT$ value with standard deviation, $\sigma_{LT}$. The parameters $\sigma_{Ct}$, and $\sigma_{LT}$ were assumed to have uninformative constant prior distributions. Parameters $\beta_0$, $\beta_1$, and $\beta_2$ were similarly given uninformative constant priors: $\beta_0$ was limited to be strictly positive; $\beta_1$ was limited to the range −100 to 0; and $\beta_2$ was limited to the range 0–30.

### Estimating the duration of infectiousness

Posterior model estimates of the relationship between Ct value and log-titre of infectious virus found a critical threshold Ct value $\beta_2$ at which the log-titer of infectious virus rapidly increased from 0 to a maximum. Under the assumption that a high log-titer of infectious virus in milk samples corresponds to an animal being infectious, we estimate the duration and timing of infectiousness by estimating the duration in which individuals in a population have a Ct value below the critical threshold value.

For each set of population parameters (means and standard deviations) from the posterior distribution of the Ct value trajectory model, we randomly draw a set of individual level parameters and simulate an individual's Ct value trajectory (1,000 trajectories simulated for each set of parameters). For all individuals we calculate the duration the Ct value remains below the critical threshold value for each value of $\beta_2$ in the posterior distribution (log-titer of infectious virus to Ct value model). This gives us the distribution for the duration of infectiousness across individuals for each set of population parameters. We calculate the mean duration of infectiousness for each set of parameters, which gives us our posterior distribution for the mean duration of infectiousness. Similarly, we can calculate the time at which 95% of individuals are no longer infectious for each set of parameters. We report the median and 95% credible intervals of each distribution in the main text.

We also estimate the posterior distribution of the mean duration of infectiousness, assuming that $\beta_2$ is known exactly to highlight how our estimate depends on the parameter value (S3 Fig). We also estimate the distribution for the duration of infectiousness using the mean values of the population parameters from the posterior distribution of the Ct value trajectory model (Fig 2C).

### Effectiveness of case isolation

As before, for each set of population parameters (mean and standard deviations) from the posterior distribution of the Ct value trajectory model, we simulate individual Ct value trajectories (1,000 trajectories simulated for each set of parameters). We then calculate the proportion of the population infectious (Ct value below threshold value, accounting for entire posterior distribution of $\beta_2$, as a function of time since infection, for each set of parameters.

We then calculate the proportion of infections prevented for different values of: the time post infection at which infected cattle are isolated (0–15 days post infection considered); and the proportion of infected cattle that are isolated (0–1 considered). We calculate the proportion of infections prevented by each pair of case isolation values for each set of population parameters giving us the posterior distribution for the proportion of infections prevented for each pair of case isolation values. We then plot the median, and central 50% and 95% credible intervals of the posterior distribution for all case isolation values considered (Fig 4).

### Supporting information

**S1 Fig. Sensitivity of Ct value trajectory model parameter estimates.** Median (point) and 95% credible intervals (lines) for posterior parameter estimates for the Ct value trajectory model using different subsets of data. Models include: fitting to all data, the main model used for analysis (dark green); fitting to all data but including an additional

parameter that adjusts the modeled Ct values between the data from the two different experimental infection studies (orange); fitting only to the data for experimentally infected cattle (purple); fitting only to the data for naturally infected cattle (pink); and fitting to all data, but allowing separate distributions for the viral decay rate between experimentally and naturally infected cattle (green for experimentally infected cattle, yellow for naturally infected cattle). Note that because naturally infected cattle are generally detected post peak the model fit to data from naturally infected cattle only estimated the viral decay rate distribution. The data underlying this figure can be found in S1 Data and at http://dx.doi.org/10.5281/zenodo.17604863.
(TIFF)

**S2 Fig. Posterior distributions of Ct value model population parameters.** Posterior samples (black points) from the Ct value model for all population parameters (means and standard deviation). The distribution of posterior samples is shown for each pair of population parameters. Also shown is the median of the posterior distribution for each individual parameter (red lines). These median parameter values are also shown in SFig.1 and were used in Fig.2a to plot the 95% interval of the estimated population distribution of Ct value trajectories (using median parameter estimates). The data underlying this figure can be found in S1 Data and at http://dx.doi.org/10.5281/zenodo.17604863.
(TIFF)

**S3 Fig. Posterior distribution for the mean duration of infectiousness.** The posterior distributions for estimates of the mean duration for which cattle's Ct value remains below a threshold value. Shown are the posterior distributions for different choices of the threshold value (colors). Also shown is the posterior distribution using the empirical distribution of the threshold value (as estimated using the model linking Ct value and the log-titer of infectious virus, 2 parameter). Ct values above the estimated threshold value represent little-to-no infectious virus and so are unlikely to be infectious. This posterior distribution thus represents our posterior distribution for the duration of infectiousness. The data underlying this figure can be found in S1 Data and at http://dx.doi.org/10.5281/zenodo.17604863.
(TIFF)

**S4 Fig. Proportion of infections prevented by case isolation.** The proportion of transmission events prevented (color) as a function of: the time post infection at which infected cattle are isolated (x-axis); and the proportion of infected cattle that are isolated (y-axis). The proportion of transmission events prevented is calculated across the entire posterior distribution for both models; plotted is the median value (50 percentile panel), and the central 50% (25–75 percentile panels) and 95% (2.5–97.5 percentile panels) credible intervals. Contour lines (proportion of transmission prevented: 0.1–0.9 in intervals of 0.1) connect points on each panel with the same value (black lines). The data underlying this figure can be found in S1 Data and at http://dx.doi.org/10.5281/zenodo.17604863.
(TIFF)

**S5 Fig. Expected Ct values of pooled milk samples with reduced milk output for infected animals.** Expected Ct value of a pooled milk sample (color) as a function of: the Ct value of the infected animal (y-axis, one infected animal contributing to the pooled sample); the total number of cattle contributing to the sample (x-axis); the efficiency of the rt-PCR test used (panels); and the milk output of the infected animal relative to uninfected animals. Note that an rt-PCR test of 100% efficiency doubles the viral concentration with each round of amplification (Ct value increasing by 1). Contour lines (Ct values: 25, 30, 35, 40) connect points on the graph with the same value (labeled black lines). The median (solid line) and 95% credible interval (dashed lines) for the population mean minimum Ct value (P) for a single infected animal is highlighted (cyan). The 95% credible interval (dashed lines) for the population distribution of minimum Ct values using mean population parameters, N(P, P), for a single infected animal is also highlighted (blue). The data underlying this figure can be found in S2 Data.
(TIFF)

                                                                  

**S1 Table. Summary of parameters in the Ct value trajectory model and their assumed prior distributions.**
(XLSX)

**S1 Data. Underlying data for modeled estimates in Figs 1–3 and S1–S4.**
(XLSX)

**S2 Data. Underlying data for Figs 4 and S5.**
(PARQUET)

## Acknowledgments

The authors wish to thank Steven Moratti for helpful discussions around dairy farming practices.

## Author contributions

**Conceptualization:** Oliver Eales, James M. McCaw, Freya M. Shearer.

**Data curation:** Oliver Eales.

**Formal analysis:** Oliver Eales.

**Funding acquisition:** Oliver Eales, James M. McCaw, Freya M. Shearer.

**Software:** Oliver Eales.

**Supervision:** James M. McCaw, Freya M. Shearer.

**Writing – original draft:** Oliver Eales.

**Writing – review & editing:** Oliver Eales, James M. McCaw, Freya M. Shearer.

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
