## [Editor Report · Decision Letter 0]

7 Jul 2025

Dear Dr Eales,

Thank you for submitting your manuscript entitled "Viral kinetics of H5N1 infections in dairy cattle" for consideration as a Research Article by PLOS Biology. My sincere apologies for the extraordinary delay.

Your manuscript has now been evaluated by the PLOS Biology editorial staff, as well as by an academic editor with relevant expertise, and I am writing to let you know that we would like to send your submission out for external peer review as a Short Report.

However, before we can send your manuscript to reviewers, we need you to complete your submission by providing the metadata that is required for full assessment. To this end, please login to Editorial Manager where you will find the paper in the 'Submissions Needing Revisions' folder on your homepage. Please click 'Revise Submission' from the Action Links and complete all additional questions in the submission questionnaire. Please, when adding the rest of the metadata choose "Short Report".

Once your full submission is complete, your paper will undergo a series of checks in preparation for peer review. After your manuscript has passed the checks it will be sent out for review. To provide the metadata for your submission, please Login to Editorial Manager (https://www.editorialmanager.com/pbiology) within two working days, i.e. by Jul 09 2025 11:59PM.

Kind regards,

Melissa

Melissa Vazquez Hernandez, Ph.D.

Associate Editor

PLOS Biology

---

## [Decision Letter · Decision Letter 1]

25 Aug 2025

Dear Dr Eales,

Thank you for your patience while your manuscript "Viral kinetics of H5N1 infections in dairy cattle" was peer-reviewed at PLOS Biology. It has now been evaluated by the PLOS Biology editors, an Academic Editor with relevant expertise, and by three independent reviewers.

In light of the reviews, which you will find at the end of this email, we would like to invite you to revise the work to thoroughly address the reviewers' reports. While the relevance of the work is noted by the reviewers, there is a share concern regarding overstatements on the conclusions. Reviewer 1 thinks the topic is relevant but that the conclusions should be temper and thinks that experimental data challenge the model outputs. Specifically, the reviewer notes that viral kinetics modeled from experimental infections may not reflect natural exposures, the dataset is small, and the pooled milk detection model may overestimate sensitivity by assuming uniform milk contributions despite reduced output from sick cows. This reviewer suggests benchmarking against additional datasets, considering dose–response effects, refining milk pooling assumptions, tempering Ct–infectivity claims, accounting for low-dose infectivity, and acknowledging immune dynamics to strengthen the manuscript. Reviewer 2 thinks that the conclusions are not representative of the complexity of the current H5N1 outbreak, so the analysis is limited. As Reviewer 1, this reviewer also says that conclusions about Ct thresholds and infectious virus shedding should be tempered by consideration of host immune responses, and also mentions the limitation of the small dataset. The reviewer suggests that you include recent studies like that of Facciuolo et al, something also mentioned by Reviewer 1. Reviewer 3 gives several recommendations such as the addition of disclaimers to prevent overinterpretation and to more explicitly connect pooled surveillance testing with subsequent isolation-based interventions.

IMPORTANT: after discussion with the Academic Editor and the reviewers, we require that you incorporate the data from the most recent studies cited by the reviewers in order to increase the sample size and demonstrate that your conclusions are robust, and to tempre your conclusions appropiately given all the caveats mentioned by the reviewers.

Given the extent of revision needed, we cannot make a decision about publication until we have seen the revised manuscript and your response to the reviewers' comments. Your revised manuscript is likely to be sent for further evaluation by all or a subset of the reviewers.

**IMPORTANT - SUBMITTING YOUR REVISION**

*Re-submission Checklist*

*Published Peer Review*

*PLOS Data Policy*

*Blot and Gel Data Policy*

Sincerely,

Melissa

Melissa Vazquez Hernandez, Ph.D.

Associate Editor

PLOS Biology

REVIEWERS' COMMENTS

Reviewer #1:

Summary

This manuscript by Eales et al. presents a timely modeling effort to estimate viral kinetics, infectiousness, and testing sensitivity for H5N1 virus in dairy cattle. The topic is relevant given the ongoing spread of H5N1 in livestock and the urgent need for evidence-based surveillance and control strategies. The authors leverage limited available data to develop their models and highlight the importance of national surveillance efforts. However, several key assumptions warrant further clarification or tempering of conclusions. Additionally, emerging experimental data challenge aspects of the model outputs, raising questions about the robustness and biological relevance of some inferences.

Major Comments

1. The authors model Ct value trajectories in milk based on published data from experimentally and naturally infected cattle. However, the infectious inoculum used in experimental infections likely differs from natural exposures, both in dose and route. Prior studies (e.g., Lee et al., https://www.researchsquare.com/article/rs-6900680/v1) have demonstrated that viral kinetics, including the timing and magnitude of peak viral load, can be influenced by infectious dose. Can the authors address how such differences might influence the trajectory parameters in their model?

2. A recent study by Facciuolo et al. (PMID: 40247094) also characterized viral growth kinetics in experimentally infected dairy cattle using milk samples. Given the limited sample size in the present study (Fig. 1A), the authors should consider incorporating or comparing their estimates with those reported by Facciuolo et al. to strengthen the robustness of their presented dataset.

3. In Figure 3, the authors model the expected Ct value of pooled milk samples and conclude that a single index case could be detected in large dairy herds (30,000-40,000 cattle) through milk vat testing. However, it is unclear how the model accounts for milk volume contributions from infected versus uninfected animals. Several cited studies report that clinically infected cattle exhibit marked reductions in milk production—particularly prior to or near peak viral loads. If a symptomatic cow contributes little or no milk to the bulk tank at the time of peak viral shedding, this would substantially reduce the effective viral concentration in pooled milk. Can the authors clarify whether this was incorporated into their calculations, or whether assumptions of uniform milk contribution may overestimate detection sensitivity?

4. The data presented in Figure 2B suggest a relationship between Ct value and infectious virus titer, with a rapid decline in recoverable infectious virus centered around a Ct value of 21.5. However, this finding, based on a single study, may overstate the existence of a strict infectivity threshold. Influenza viruses—including H5N1—have been isolated from primary samples with Ct values above this range. Given the cross-sectional nature of the data and the lack of longitudinal measurements from individual animals, the model may not adequately reflect the true variability in infectious virus recovery. For example, in a report by Halwe et al., a relationship between Ct value (Fig. 3C) and infectious titer (Fig. 4A) demonstrates that Ct values >21 can still yield infectious virus. The authors should consider either supporting this relationship with additional experimental data or tempering their conclusions accordingly.

5. The infectiousness model presented in Figure 2C would benefit from additional context based on recent experimental findings. Lee et al. (https://www.researchsquare.com/article/rs-6900680/v1) showed that as little as 10 TCID50 of virus was sufficient to establish infection in the bovine mammary gland, suggesting that infectious dose thresholds may be lower than assumed in the model. This raises the possibility that animals with Ct values above the modeled "critical threshold" could still contribute to transmission.

6. The estimate that 95% of cattle are no longer infectious by 16.6 days post-infection does not appear to account for host immune responses. While viral RNA may persist in milk for some time (>10 days), the presence of neutralizing antibodies—especially in recovered animals—is likely to substantially reduce or eliminate the potential for onward transmission. For example, Halwe et al. report difficulty isolating virus beyond 9 days post-infection, despite RNA-positive milk, likely due to neutralizing antibodies. The 16.6-day estimate may overstate infectious potential in natural settings. Incorporating immune dynamics, or at least acknowledging this limitation, would strengthen the interpretation of the infectious period.

Minor Comments

Please double-check citation numbering. Caserta et al. is labeled as "[4]" in the Methods section but appears as "[3]" in the reference list.

Reviewer #2:

In the manuscript by Eales et al, the authors seek to quantify epidemiological parameters associated with H5N1 outbreaks on dairy farms through analysis of published data from 3 studies (2 experimentally infected and 1 naturally infected). The authors use published data to suggest that infectious virus would only be observed in milk samples with a Ct value less than or equal to 21.5 and that the average duration for shedding of infectious virus of 6.2 days post infection. The authors also model the theoretical sensitivity of qPCR to detect a single H5N1 positive animal and conclude that this is feasible within 30,000 cows if a single cow has a CT value <25. While the conclusions are reasonable based on the data used to generate the model, they are not representative of the complexity of the ongoing H5N1 outbreak on dairy cattle. Thus, the utility of the analysis is significantly limited.

Major comments:

1. The authors have a limited number of cows in the analysis - 5 experimentally infected cows and 12 naturally infected cows (from a single farm). Based on the graphs presented in the manuscript the kinetics of shedding seem heterogenous in the naturally infected animals compared to the experimentally infected animals. It is unclear how similar the kinetics are across the different infection conditions or across animals in the naturally infected group. Addition of more animals could help refine the different infection kinetics that are present during an outbreak. Additional publications to include: (Facciuolo et al, Nature Microbiology; Faccin, F.C. et al. doi.org/10.21203/rs.3.rs-6915991/v1, Lee, C. et al. doi.org/10.21203/rs.3.rs-6900680/v1)

2. Because the analyses in this paper are almost all based on the idea that Ct value can be linked to infectious virus, it is critical to note properties that can influenza infectious titer in milk. Specifically, this manuscript makes no mention of induction of host immune responses and the role they may play in the duration of infectiousness. In each of the 2 controlled infection model studies used for this analysis, antibody titers are tracked over time, thus it is known when post-infection antibody levels started to rise. A raise in antibodies within the milk as infectivity drops is an important consideration for CT to infectious titer analysis. In Baker et al, infectious virus is isolated at 12dpi but not after and that cow had a virus neutralization titer of 160 by that day; in each cow, neutralizing antibody titers had started to rise by 9-10dpi, respectively. In Halwe et al antibody levels rise ~ 7-10 days post inoculation. Since these datasets are used in these analysis reported, it is an oversight not to mention that these antibody responses likely relate to lack of recovery of infectious virus.

Specific comments:

1. The manuscript title is misleading as there is no novel data on the viral shedding. Please revise to be more representative of the analysis conducted.

2. It might help the organization and flow of the manuscript if the 'Pooled testing of milk vats' section was moved to just before the 'Limitations' section. Currently, subpanels in the figures are discussed out of order which could be avoided by moving the 'pooled milk' section to the end. Rearranging this would also allow all analyses of samples from individual cows (Ct value trajectories, Ct x log titer, duration of infectiousness, and the discussion of isolation efficiency) to be discussed together, which seems like a more logical flow.

3. Details are insufficient for the following areas and additional references are needed:

a. National Milk Testing Strategy (NMTS) - is wide spread use, strategy, limitations, and achievements.

b. Impact of clinical vs subclinical cows is examined in Peña-Mosca et al, 2025, Nature Communications, and should be included and discussed.

Minor comments:

1. The authors need to carefully read over the manuscript.

a. Edit references to 'high' and/or 'low' Ct values as appropriate. In a couple places the manuscript reads 'high' Ct values when the context of the sentence leads me to think 'low' Ct values (high viral load) was intended, and vice versa. The abstract has a couple instances of this, as well as the penultimate sentence of Introduction.

b. Check for misspellings - one example: figure one: 'free' should be 'three'.

Reviewer #3:

The manuscript "Viral kinetics of H5N1 infections in dairy cattle" has been submitted to PLOS Biology by Eales, McCaw, and Shearer. In this paper, the authors assemble three sources of H5N1 kinetics data from dairy cattle. Two sources are experimental (mammary inoculation), and the other is observational. The paper proceeds logically by first developing a modeling approach that learns a posterior distribution of viral kinetics from all three data sources, considers the sensitivity of pooled testing, and then assembles the estimates required to evaluate the effectiveness of isolating infected cattle.

This is a valuable paper because it integrates data across studies to provide valuable estimates of the effectiveness of surveillance, the infectiousness/Ct relationship, and the potential effectiveness of isolation. The conclusions follow from the data, the limitations are appropriate, and the writing is clear. I support this paper's publication after minor revisions. Below, I provide some constructive suggestions.

In the analysis of Duration of Infectiousness, the authors provide estimates for the proportion of animals that do not become infectious, the upper 2.5% of infection durations, and the 95% credible interval for the duration of infectiousness. These values are estimated from a small sample size of mostly inoculated cattle, using distributions chosen for modeling convenience (but not because their tails are necessarily correctly specified). Therefore I recommend that these sorts of estimated tail values include an appropriate disclaimer to caution readers against overinterpretation.

The paper could also benefit from fleshing out the connection between pooled surveillance testing and cattle screening and isolation. Would it even be plausible to link surveillance and isolation-based interventions to suppress outbreaks quickly?

For instance, the authors make the convincing case that weekly pooled testing is likely to detect outbreaks. And, they explore the proportion of transmission prevented by case isolation (particularly in Fig 4B). What happens when we put them together? What are the implications of the timing of detection via surveillance for the subsequent possible interventions? Is the first wave of secondary infections likely to be large because of a delay in detection, even if effective screening and isolation is possible? Does the delay between infection and symptoms present additional challenges to effective isolation? Rather than quantitative analyses that invoke additional assumptions these issues could be commented on, or discussed in the limitations -- which additional parameters would need to be nailed down before it would be possible to perform control analyses e.g. Kissler et al (PLOS Bio 2021) or Middleton et al (Sci Adv 2024)?

Finally, I was intrigued to see the acknowledgement of Steven Moratti about helpful discussions around dairy farming practices. There were a few references to typical dairy practices throughout the paper, and I admit that my curiosity was piqued each time. To the extent that the authors and editor feel it would be appropriate, I would suggest a brief explanation in the Introduction of typical dairy practices, the route of H5N1 transmission, the natural history of infection, and common interventions. My hope would be that even a few sentences here would help readers to understand the context of the modeling, particularly given the disproportionate focus of the infectious disease modeling community on human hosts.

---

## [Decision Letter · Decision Letter 2]

10 Dec 2025

Dear Dr Eales,

Thank you for your patience while we considered your revised manuscript "Viral kinetics of H5N1 infections in dairy cattle" for publication as a Short Reports at PLOS Biology. This revised version of your manuscript has been evaluated by the PLOS Biology editors, the Academic Editor and one of the original reviewers.

Based on the reviews and on our Academic Editor's assessment of your revision, we are likely to accept this manuscript for publication, provided you satisfactorily address the remaining points raised by Reviewer 1. Please also make sure to address the following data and other policy-related requests.

1) We routinely suggest changes to titles to ensure maximum accessibility for a broad, non-specialist readership, and to ensure they reflect the contents of the paper. In this case, we would suggest a minor edit to the title, as follows. Please ensure you change both the manuscript file and the online submission system, as they need to match for final acceptance:

"Modelling of H5N1 influenza virus kinetics during dairy cattle infection suggests how long animals remain infectious"

2) Please update your abstract to says that you have look to 4 studies, instead of the original 3.

3) Please add the links of the funding agencies in the Financial Disclosure statement in the manuscript details.

4) Please cite the location of the data clearly in all relevant main and supplementary Figure legends, e.g. “The data underlying this Figure can be found in S1 Data” or “The data underlying this Figure can be found in https://doi.org/10.5281/zenodo.XXXXX”

5) Supplementary files (e.g., excel). Please ensure that all data files are uploaded as 'Supporting Information' and are invariably referred to (in the manuscript, figure legends, and the Description field when uploading your files) using the following format verbatim: S1 Data, S2 Data, etc. Multiple panels of a single or even several figures can be included as multiple sheets in one excel file that is saved using exactly the following convention: S1_Data.xlsx (using an underscore).

We expect to receive your revised manuscript within two weeks.

*Published Peer Review History*

*Press*

Sincerely,

Melissa

Melissa Vazquez Hernandez, Ph.D.

Associate Editor

PLOS Biology

REVIEWER 1'S COMMENTS

Reviewer #1:

This revised submission by Eales et. al. was well responsive to this reviewer's comments. New data was incorporated, in line with newly published studies (increasing robustness). I appreciate the assessment of new parameters (e.g., effects of reduced milk output on pooled testing). Further, the authors delved deeper into potential limitations of their study, which further serve to balance their modeling assumptions with more complex biological associations. This reviewer is satisfied with the manuscript in its current form. However, some very minor points could be addressed.

Minor:

Consider updating the introduction regarding the current state of H5N1 in the U.S.: "As of 21 May 2025, 1070 cases…"

* These aren't truly "cases" as individual testing is not done. Rather, this is "herd-level" surveillance.

* Now in December 2025, the number of positive herds and total number of states affected has changed. Consider updating to reflect the most recent data.

The authors note "…one study that inoculated a single cow with significantly lower volume of virus…" needs correcting. In the report by Lee et. al., four cows were infected, with each cow receiving three different infectious doses that varied by quarter. Please revisit the experimental setup of this study and revise accordingly.

---

## [Editor Report · Decision Letter 3]

17 Dec 2025

Dear Dr Eales,

Thank you for the submission of your revised Short Reports "Modelling of H5N1 influenza virus kinetics during dairy cattle infection suggests the timing of infectiousness" for publication in PLOS Biology. On behalf of my colleagues and the Academic Editor, Jason T. Ladner, I am pleased to say that we can in principle accept your manuscript for publication, provided you address any remaining formatting and reporting issues. These will be detailed in an email you should receive within 2-3 business days from our colleagues in the journal operations team; no action is required from you until then. Please note that we will not be able to formally accept your manuscript and schedule it for publication until you have completed any requested changes.

IMPORTANT: I wanted to let you know that PLOS will be "closed" from December 22nd to January 3rd, and therefore some responses will be slow. We thank you for your patience!

PRESS

Sincerely, 

Melissa

Melissa Vazquez Hernandez, Ph.D., Ph.D.

Associate Editor

PLOS Biology
